# Anti-Obesity and Anti-Diabetic Effects of *Ishige okamurae*

**DOI:** 10.3390/md17040202

**Published:** 2019-03-29

**Authors:** Hye-Won Yang, K.H.N. Fernando, Jae-Young Oh, Xining Li, You-Jin Jeon, BoMi Ryu

**Affiliations:** Department of Marine Life Science, Jeju National University, Jeju 63243, Korea; koty221@naver.com (H.-W.Y.); hiruninfdo@gmail.com (K.H.N.F.); ojy0724@naver.com (J.-Y.O.); xiningmarinesci666@naver.com (X.L.)

**Keywords:** *Ishige okamurae*, marine alga, obesity, diabetes, nutraceuticals

## Abstract

Obesity is associated with several health complications and can lead to the development of metabolic syndrome. Some of its deleterious consequences are related to insulin resistance, which adversely affects blood glucose regulation. At present, there is a growing concern regarding healthy food consumption, owing to awareness about obesity. Seaweeds are well-known for their nutritional benefits. The brown alga *Ishige okamurae* (IO) has been studied as a dietary supplement and exhibits various biological activities in vitro and in vivo. The bioactive compounds isolated from IO extract are known to possess anti-obesity and anti-diabetic properties, elicited via the regulation of lipid metabolism and glucose homeostasis. This review focuses on IO extract and its bioactive compounds that exhibit therapeutic effects through several cellular mechanisms in obesity and diabetes. The information discussed in the present review may provide evidence to develop nutraceuticals from IO.

## 1. Introduction

Over the past 50 years, obesity has become a global public health issue that negatively affects quality of life and increases the risk of various illnesses and healthcare costs worldwide [1,2,3]. Obesity is considered a risk factor for coronary artery diseases, cerebrovascular accidents, type-2 diabetes mellitus, systemic hypertension, various cancers, fatty liver disease, osteoarthritis, and gynecological disorders [4]. An understanding of the molecular basis of obesity-associated diseases is required to approach its prevention. The properties of adipose tissue and adipocytes in obesity have been studied [5], and Higdon and Frei [6] also emphasized that obesity is a chronic oxidative stress condition due to an imbalance among tissue active oxygen, reactive oxygen species (ROS) and antioxidants.

Oxidative stress also plays a key role in the pathogenesis of many other progressive diseases including diabetes, atherosclerosis and cancer [7,8,9]. In addition, lipid accumulation has been correlated with various markers of systemic oxidative stress [10]. Furukawa et al. [11] reported that oxidative stress mediates the obesity-associated development of metabolic syndrome via two mechanisms: (1) increased oxidative stress due to lipid accumulation leads to dysregulated production of adipocytokines, and (2) selective increase in ROS production due to lipid accumulation leads to elevation in systemic oxidative stress. Oxidative stress can activate a series of stress pathways involving a family of serine/threonine kinases, resulting in a negative effect on insulin signaling [12], and an increase in the production of free radicals or impaired antioxidant defenses. Diabetes is characterized by hyperglycemia and insufficiency in the secretion or action of endogenous insulin [13]. An increase in oxidative stress can lead to hyperglycemia in both type-1 and type-2 diabetes [14,15]. Tan et al. [16] showed that hydrogen peroxide (H_2_O_2_) stimulated the inhibition of insulin-induced glucose uptake in vitro. In that study, oxidative stress directly causes insulin resistance via overactivation of extracellular signal-related kinase (ERK).

Current available therapies for obesity and diabetes have either limited efficacy or cause side effects. Therefore, many studies have suggested that natural sources can be used as complementary treatments and preventive materials with less toxic and fewer side effects [17]. Marine algae have been identified as rich sources of structurally diverse bioactive compounds including pigments, fucoidans, phycocolloids, and phlorotannis, with nutraceutical and biomedical potential [18,19]. *Ishige okamurae* (IO) is an edible brown seaweed found in temperate coastal areas, such as the Korean peninsula [20]. It is abundant along the coast of Jeju Island and is a potential functional food. In this review, we discuss the anti-obesity and anti-diabetic effects of IO extract and its cellular mechanisms of action. We also suggest its use as a potential nutraceutical source.

## 2. Anti-Obesity and Anti-Diabetic Properties of IO Extract

*Ishige* is a genus of brown algae with two species—*Ishige foliacea* and IO. Studies on the extracts of *Ishige*, including *Ishige foliacea* and IO, have reported various in vitro and in vivo activities, such as antioxidant, anti-diabetic, and anti-obesogenic effects [21,22,23]. Owing to these bioactivities, these extracts have been gaining increased attention in recent years for potential nutraceutical application in metabolic syndrome.

Metabolic syndrome is characterized by an increase in ROS levels, which cannot be counteracted by endogenous antioxidant systems [24]. The increase in ROS levels plays a key role in the development of metabolic diseases, which could lead to changes in glucose uptake, exacerbating diabetes mellitus, obesity, cardiovascular diseases, or cancer [24,25]. The antioxidant properties of *Ishige foliacea* and IO methanol extracts have been investigated in terms of their free-radical which includes 1,1-diphenyl-2-picryl hydrazyl (DPPH); 2,2-azobis(3-ethylbenzothiazoline-6-sulfonate (ABTS) and nitrite scavenging activity [21]. Furthermore, Heo and Jeon [26] reported that IO enzymatically extracted with different carbohydrases and proteases exhibits antioxidative effects. In particular, Ultraflo extract, which is a carbohydrase-based enzymatic extract, can scavenge free radicals, and Kojizyme extract, which is a protease-based extract, can reduce the DNA damage caused by hydrogen peroxide (H_2_O_2_). Thus, the antioxidant efficacy of IO extracts indicates that it is a potential functional food, which can be used as a supplement for patients with metabolic syndrome.

Diabetes and obesity are common, closely interrelated disorders and are caused by poor metabolic conditions. Obesity, a characteristic feature of metabolic syndrome, involves the accumulation of abnormal or excessive fat that may interfere with the maintenance of an optimal state of health [27]. It is also associated with a systemic increase in oxidative stress, resulting in adipokine imbalance [28]. Previous in vitro and in vivo studies have suggested that oxidative stress can cause obesity through increased proliferation of pre-adipocytes and increased size of differentiated adipocytes [29,30].

Cha and Cheon [31] showed that IO extract can inhibit lipid accumulation induced during adipogenesis of 3T3-L1 preadipocytes. The concerted regulation of gene expression by various adipogenic factors is required for the differentiation of preadipocytes to adipocytes. Furthermore, research on the mechanism underlying preadipocyte mitogenesis and differentiation into adipocytes may help understand the initiation and progression of obesity and its associated diseases. Peroxisome proliferator-activated receptors (PPARγ) has been studied for its involvement in the regulation of nutrient sensing and glucose and lipid metabolism [32]. Expression level of PPARγ is highest in adipose tissue [33] when it regulates the transcriptional cascade involved in adipocyte differentiation [34]. The hormone nuclear receptor PPARγ plays an important role in the regulation of downstream adipogenic genes [33]. Expression levels of PPARγ mRNA were significantly decreased by the IO extract during 10 days of induction [31]. Thus, the antioxidant effect of IO extract can inhibit the accumulation of lipids and modulate PPARγ expression. Although IO extract can decrease the levels of PPARγ, previous studies showed the effect of IO extract against obesity through other adipogenic transcription factors. The IO extract can suppress the increase in lipid droplet size by reducing the expression of adipogenic transcription factors in white adipose tissue (WAT), which is larger in a high-fat diet (HFD)-fed mice than in mice on a normal diet [35,36]. Therefore, IO extract can reduce body weight gain by preventing an increase in WAT mass and ameliorating HFD-induced obesity.

Adipose tissue helps maintain glucose and lipid homeostasis through the secretion of various factors and through neural networks [37,38,39]. Diabetes in obese people occurs mostly due to insulin resistance and subsequent hyperinsulinemia through adipogenesis and the insulin signaling pathway. In addition, oxidative stress has been linked with disruption of insulin secretion by pancreatic β-cells [40], glucose transport in muscle [8], and adipocytokines [41].

A widely used preclinical model of diabetes is *db*/*db* mice, characterized by hyperglycemia, hyperinsulinemia, hyperleptinemia, and obesity, similar to type-2 diabetes [42]. C57BL/KsJ-*db*/*db* mice were fed a standard semi-synthetic diet (AIN-93G) with IO extract (0.5%, w/w; IO extract supplementation), resulting in downregulated fasting blood glucose levels. IO extract supplementation controlled blood glucose levels during the intraperitoneal glucose tolerance test (IPGTT) [22]. Homeostatic model assessment (HOMA) is a method for assessing insulin resistance (IR) and is a useful index of insulin sensitivity [43]. It has been shown that HOMA-IR is lowered following IO extract supplementation [22]. Previous studies have suggested that therapeutic agents may be required to prevent hyperglycemic conditions in patients with early-stage type-2 diabetes. IO extract supplementation significantly lowered glycated hemoglobin (HbA1c) levels [22], which is useful for monitoring glycemic control in diabetic patients [44]. Taken together, IO extract supplementation can control blood glucose levels and improve insulin resistance in *db*/*db* mice. We suggest that IO extract can be used as an antidiabetic supplement.

## 3. Composition of IO

Many brown algae species are used as food ingredients and supplements and possess a variety of biological activities. These biological activities are related to the presence of polyphenols, polysaccharides and pigments. Among polyphenols, one of the most common classes of secondary metabolites derived from polymerized phloroglucinol units are phlorotannins [45]. Phlorotannins are tannin derivatives composed of several phloroglucinol units isolated from brown algae [19]. It has been reported that brown algae are richer in phlorotannins than other marine algae. Polyphenols can react with oxidants in one-electron reactions, pairing with the free electron of the oxidant to become chemically inactive. Therefore, polyphenols act as antioxidants that inhibit the formation of free radicals in biological systems [46]. In addition, previous studies have examined various biological activities associated with polyphenols from brown algae, including antioxidant, anti-coagulant, anti-bacterial, anti-inflammatory, and anti-cancer effects [18,47,48]. Thus, phlorotannins isolated from brown seaweeds represent the most widely studied class of secondary metabolites in marine organisms, with potential use in the nutraceutical and functional food industry.

Yoon et al. [49] studied the secondary metabolites of IO extract with antioxidant effects, including phloroglucinol, 6,6′-bieckol, and diphlorethohydroxycarmalol (DPHC). Octaphlorethol A (OPA) was also isolated and purified from IO extract [50]. Recently, a novel polyphenol-compound, ishophloroglucin A (IPA) with α-glucosidase inhibitory activity was isolated from IO extract [51]. Zou et al. [52] evaluated the antioxidant effects of 6,6′-bieckol and DPHC by using the electron spin resonance (ESR) technique. The two phlorotannins displayed potent radical scavenging activities against DPPH as well as hydroxyl, alkyl, and superoxide radicals. Moreover, effective concentration (EC_50_) values of phlorotannins, defined as the concentration at which the radicals were scavenged by 50%, are summarized in Table 1. Heo and Jeon [53] reported the cytoprotective effect of DPHC in Vero cells against oxidative stress induced by hydrogen peroxide (H_2_O_2_).

Several studies have reported that among the pigments of brown algae, fucoxanthin can reduce oxidative stress and symptoms of metabolic syndrome via its anti-diabetic and anti-obesogenic effects [54,55,56,57]. In addition, Kang et al. [58] reported that fucoxanthin isolated from IO can reduce high glucose-induced oxidative stress in human umbilical vein endothelial cells (HUVEC) and in zebrafish models. Taken together, the antioxidative effects of IO may be effective as supplementary treatment of metabolic syndrome, including obesity and diabetes.

### 3.1. Anti-Obesity Effect of IO

When caloric expenditure is lower than caloric intake, adipocytes play a critical role by storing triacylglycerol and regulating metabolism in obesity. In fat tissue, adipocyte differentiation and lipid accumulation occur through adipocyte-specific proteins including enhancer binding protein (C/EBP), sterol-regulatory element-binding protein 1c (SREBP-1c), peroxisome proliferator activated receptor-γ (PPARγ), adiponectin, perilipin, fatty acid synthase (FAS), fatty acid binding protein (FABP4), and leptin [59]. According to Cha and Cheon [31], IO extract is known to inhibit lipid accumulation, which is induced during adipogenesis from 3T3-L1 preadipocytes. Previous studies have focused on the inhibition of lipid accumulation in 3T3-L1 cells through decreased expression levels of adipogenic-specific factors by polyphenols such as dieckol [60], epigallocatechin-3-gallate [61], and resveratrol [62]. Several studies have found that IO extract inhibited fat accumulation in 3T3-L1 cells through a molecular mechanism involving adipocyte-specific proteins.

DPHC from IO extract has potential antiadipogenic effects elicited via the inhibition of adipocyte differentiation and adipogenesis. Kang et al. [63] reported that levels of the adipogenesis-specific proteins including C/EBPα, SREBP-1c, PPARγ, and adiponectin were decreased to activate molecular mechanisms involved in 3T3-L1 adipocyte differentiation. These transcription factors are highly expressed in adipocytes and are involved in the mediation of lipid synthesis, lipolysis, and glucose uptake in adipocytes. DPHC can disrupt fatty acid synthesis by downregulating adipocyte-specific proteins including perilipin, FAS, FABP4, and leptin. Furthermore, DPHC can activate adenosine monophosphate-activated protein kinase (AMPK) and acetyl-CoA carboxylase (ACC), resulting in the inhibition of lipogenesis, adipocyte differentiation, and fatty acid synthesis in adipocytes. Besides the increase in AMPK and ACC, preadipocyte apoptosis also has an anti-obesity effect. Park et al. [64] reported that DPHC induced apoptosis in 3T3-L1 preadipocytes through the intrinsic pathway by regulating the protein levels of Fas, Bax, Bcl-2, caspase-9, caspase-3, and PARP. Taken together, DPHC can be used as a potential therapeutic agent against obesity.

### 3.2. Anti-Diabetic Activity of IO

Diabetes, a serious chronic metabolic disease, may develop with obesity and ageing in the general population. In addition, rapidly increasing blood glucose levels are a result of the hydrolysis of starch by pancreatic α-amylase and glucose uptake by intestinal α-glucosidases. These enzymes play a crucial role in the effective regulation of glucose absorption [65]. Therefore, an important strategy for suppressing postprandial hyperglycemia is the inhibition of α-amylase and α-glucosidase activities [66,67]. A previous study with C57BL/KsJ-*db*/*db* mice showed that IO extract supplementation prevented insulin resistance and regulated blood glucose levels in hyperglycemia [22]. Lee and Jeon [19] focused on developing potential anti-diabetic nutraceutical and functional foods from phlorotannins.

Furthermore, several studies found that IO extract showed anti-diabetic activity by inhibiting α-amylase and α-glucosidases. Phlorotannins isolated from IO extract have excellent anti-diabetic properties. DPHC (IC_50_ = 0.53 ± 0.08 and 0.16 ± 0.01 mM) showed effective inhibitory effects against α-amylase and α-glucosidase compared to acarbose (IC_50_ = 1.10 ± 0.07 and 1.05 ± 0.03 mM), which was used as the positive control [68]. DPHC significantly suppressed the increase in postprandial blood glucose levels in both streptozotocin-induced diabetic and normal mice after the consumption of starch [68]. Moreover, Lee et al. [69] described that DPHC treatment protected high glucose-induced damage in RINm5F pancreatic β-cells. The dysfunction of pancreatic β-cells has a central role in the pathogenesis of type-2 diabetes [70]. Therefore, DPHC can delay the absorption of dietary carbohydrates and improve secretory responsiveness of insulin following stimulation with glucose.

Recently, IPA, a novel polyphenol-compound derived from IO extract showed a solid α-glucosidase inhibitory activity [51]. The study showed the application of IPA in standardizing the inhibition of α-glucosidase activity of IO extract and proposed IPA potential in the application of marine-derived nutraceuticals.

## 4. Potential Nutraceutical Use of IO

The role of food is to provide enough nutrients to meet metabolic requirements, which is relevant to well-being, good health, and disease management [71]. Recently, consumer awareness of bioactive compounds as functional ingredients has increased, and knowledge about their various health benefits is increasing. Seaweeds are rich sources of structurally diverse bioactive compounds with valuable nutraceutical, pharmaceutical and cosmeceutical potentials [72]. Antioxidant properties of seaweeds enable their use as nutraceuticals and functional food ingredients [73]. A considerable number of bioactive compounds has been isolated from seaweeds and evaluated for their potential as functional food ingredients to assist in the treatment of metabolic diseases such as cancer, hypertension and diabetes [73].

IO is an edible brown seaweed that grows on rocks in the upper and middle intertidal zones in the northwest Pacific Ocean (Korea, Japan, and China), where it forms continuous bands [20]. As shown in Table 2, previous studies have discussed the usefulness of bioactive compounds from IO as functional ingredients [49,58,74]. Additionally, the antioxidant properties of the methanol extract and enzymatic extract from IO were evaluated to develop potential functional food materials against oxidative stress [21,26]. IO extract is rich in secondary metabolites such as phlorotannins, carotenoids, and polysaccharides with various bioactive properties.

6,6′-bieckol is another phlorotannin from IO which possesses in vitro and in vivo neuroprotective effects. 6,6′-bieckol suppresses acetylcholinesterase (AChE) activity with an IC_50_ value of 46.42 ± 1.19 µM [49]. AChE plays a key role in the regulation of several physiological reactions by hydrolyzing the neurotransmitter acetylcholine in the cholinergic synapses [81,82]. In addition, Alzheimer’s disease (AD) is related to a deficit in cholinergic functions in the brain [83]. Thus, the application of 6,6′-bieckol as an alternative for AChE inhibitors suggests a therapeutic potential in AD. Fucoxanthin, an accessory pigment in chloroplasts, is a well-known brown seaweed carotenoid with numerous important bioactive properties [84]. It is also one of the major constituents of IO. Kim et al. [80] showed that fucoxanthin from IO reduced the production of nitric oxide (NO) and inflammatory mediators, including inducible nitric oxide synthase (iNOS) and cyclooxygenase-2 (COX-2), and inhibited nuclear factor (NF)-κB activation and mitogen-activated protein kinases (MAPKs; JNK, ERK and p38) signal pathways in LPS-stimulated RAW264.7. In LPS-treated macrophages, pro-inflammatory cytokines and gene expression were upregulated through NF-κB activation and MAPK signaling pathways [85,86]. Cancer is characterized by uncontrolled cell growth and spread [87]. Fucoxanthin has the potential to inhibit the proliferation of melanoma cell lines (B16F10 cells) through cell cycle arrest during the G0/G1 phase and the apoptotic pathway [64]. Fucoxanthin also decreases Bcl-xL expression level, which is a critical regulator of the apoptotic pathway. Moreover, it has been shown that fucoxanthin suppressed in vivo growth of B16F10 melanoma in Balb/c mice. Therefore, researchers have been interested in identifying new anti-cancer drugs from marine sources, which supposedly have fewer adverse side effects unlike synthetic drugs [88].

## 5. Conclusions

Obesity is associated with lipid accumulation together with oxidative stress, which increases insulin resistance and eventually results in diabetes. In this review, we have discussed the antioxidant properties of IO extract and the mechanisms of action underpinning its anti-obesity and anti-diabetic effects (Figure 1).

The significant health benefits associated with IO may represent an interesting progress in the search for novel functional applications. IO can also be used as a therapeutic agent and functional food against metabolic syndrome.

## Figures and Tables

**Figure 1 marinedrugs-17-00202-f001:**
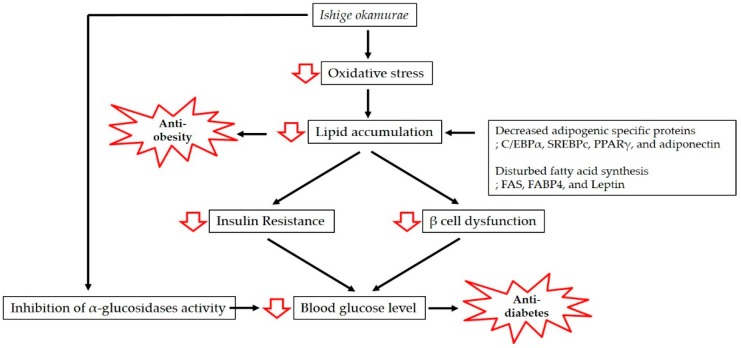
Mechanisms involved in the anti-obesity and anti-diabetic effects of *Ishige okamurae*.

**Table 1 marinedrugs-17-00202-t001:** EC_50_ values of phlorotannins from *Ishige okamurae*.

	EC_50_ (µM ± SD)			
	DPPH	Hydroxyl	Alkyl	Superoxide
6,6′-bieckol	9.1 ± 0.4	23.7 ± 1.1	17.3 ± 1.0	15.4 ± 0.9
Diphlorethohydroxycarmalol (DPHC)	10.5 ± 0.5	27.1 ± 0.9	18.8 ± 1.2	16.7 ± 0.6
Phloroglucinol	Not determined	408.5 ± 3.7	103.5 ± 1.9	124.7 ± 2.4

**Table 2 marinedrugs-17-00202-t002:** Bioactivities of functional ingredients from *Ishige okamurae*.

Functional Ingredient	Bioactivities	References
Methanolic extract	Antioxidant, anti-MMP, and anti-diabetic	[21,22,74]
Ethanolic extract	Anti-inflammatory	[75]
Enzymatic extract	Antioxidant	[26]
Fermented extract	Radioprotective and antioxidant	[76]
6,6′-Bieckol	Cholinesterase inhibition	[49]
Diphlorethohydroxycarmalol (DPHC)	Antioxidant, anti-cancer, anti-HIV, anti-obesity, and anti-diabetic	[52,53,64,68,69,77,78,79]
Fucoxanthin	Antioxidant, anti-inflammatory	[58,80]
Ishigoside	Antioxidant	[20]
Ishophloroglucin A (IPA)	α-glucosidases inhibition	[51]
Phloroglucinol	Cholinesterase inhibition	[49]

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
