# Peer review of "Anti-Obesity and Anti-Diabetic Effects of Ishige okamurae"

_marinedrugs, 2019, doi:10.3390/md17040202_

Round 1

Reviewer 1 Report

In general, this review is interesting. However, I do find that it overstates the usage of IO extract as preventive medicine for obesity and diabetes. Please tone down its medicinal benefits and use instead a terminology of a supplementary/complementary benefits throughout the text.

Line 28-30. The fact that obesity is characterized by oxidative stress does not contradict research on adipocytes in obesity. Please rephrase.

Line 43 – replace “thus” with “in that study”.

Line 45-48. “as alternatives” -> “as supplements or complementary treatment”. Please emit the end of the sentence - “than synthetic drugs for obesity and diabetes”.

Line 63 “causing” -> “exacerbating”.

Line 71 – “for the prevention” -> “as a supplement for patients with”.

Line 88-90 – PPARg drives adipogenesis but is also Important for adipose tissue health. Its inhibition is not necessarily positive, making the picture more complex. For example, TZDs, which are a class of anti-diabetic drugs, activate PPARg and increase glucose tolerance. Please discuss.

Line 99 – muscle8? Is it a reference?

Line 99-101 – please move the sentence on db/db mice to the next paragraph.

Line 102-111 - IO extract supplementation. How was it administered? In food supply or given orally via gavage?

Line 141 – “may be associated with the prevention” -> “may be effective as supplementary treatment of”

Line 158-159 –“these proteins are the downstream promoters of adipocyte-specific proteins, which are involved…” – “these transcription factors are highly expressed in adipocytes and are involved… ”

Line 175-176 - In food supply or given orally via gavage?

Line 178-179 - move last sentence to the next paragraph.

Line 188 –“ secretory responsiveness “of insulin”…”?

Line 195 – “prevention” -> “management”

Line 101 – “prevent” -> “assist in the treatment of”

Line 229-230 – “which supposedly have fewer”

Line 232 – “caused by oxidative stress, which increases” -> “together with oxidative stress, which increase”

Author Response

Dear Academic Editor,

We are gratitude for your comments and spending your valuable time to read our manuscript. We have strongly considered your suggestions and have provided the answers to each comment. The revised contents are highlighted with red color in the revised manuscript.

Comment 1

In general, this review is interesting. However, I do find that it overstates the usage of IO extract as preventive medicine for obesity and diabetes. Please tone down its medicinal benefits and use instead a terminology of supplementary/complementary benefits throughout the text.

Response

We appreciate your comment. In order to tone down medicinal benefits of IO extract in obesity and diabetes, we replace the words/sentences in manuscript.

Comment 2

Line 28-30. The fact that obesity is characterized by oxidative stress does not contradict research on adipocytes in obesity. Please rephrase.

Response

Thank you for your comment. We rewrite the sentence to “The properties of adipose tissue and adipocytes in obesity have been studied [5], and Higdon and Frei [6] also emphasized that obesity is a chronic oxidative stress condition due to an imbalance among tissue active oxygen, reactive oxygen species (ROS), and antioxidants.” (Line 28-30)

Comment 3

Line 43 - replace “thus” with “in that study”.

Response

Thank you for your comment. We replace the word to “in that study” (Line 43).

Comment 4

Line 45-48. “as alternatives” -> “as supplements or complementary treatment”. Please emit the end of the sentence – “than synthetic drugs for obesity and diabetes”.

Response

Thank you for your comment. We replace the word to “as supplements or complementary treatment”, and emit the sentence “than synthetic drugs for obesity and diabetes” (Line 46-47).

Comment 5

Line 63 “causing” -> “exacerbating”.

Response

Thank you for your comment. We replace the word to “exacerbating” (Line 62).

Comment 6

Line 71 – “for the prevention” -> “as a supplement for patients with”

Response

Thank you for your comment. We replace the word to “as a supplement for patients with” (Line 71).

Comment 7

Line 88-90 – PPARg drives adipogenesis but is also important for adipose tissue health. Its inhibition is not necessarily positive, making the picture more complex. For example, TZDs, which are a class of anti-diabetic drugs, activate PPARg and increase glucose tolerance. Please discuss.

Response

Thank you for your most valuable suggestion. We add more information about PPARγ in the section “Anti-obesity and anti-diabetic properties of IO extract” as below.

Peroxisome proliferator-activated receptors (PPARγ) has been studied to be involved in the regulation of nutrient sensing, glucose and lipid metabolism [32]. Expression level of PPARγ is highest in adipose tissue [33], and regulates the transcriptional cascade involved in adipocyte differentiation [34].” (Line 83-86).

Although IO extract can decrease the levels of PPARγ, previous studies showed that the effect of IO extract against obesity through another adipogenic transcription factors.” (Line 90-92).

32. Polvani, S.; Tarocchi, M.; Tempesti, S.; Bencini, L.; Galli, A. Peroxisome proliferator activated receptors at the crossroad of obesity, diabetes, and pancreatic cancer. World journal of gastroenterology 2016, 22, 2441.

33. Tontonoz, P.; Hu, E.; Spiegelman, B.M. Stimulation of adipogenesis in fibroblasts by PPARγ2, a lipid-activated transcription factor. Cell 1994, 79, 1147-1156.

34. Rosen, E.D.; Sarraf, P.; Troy, A.E.; Bradwin, G.; Moore, K.; Milstone, D.S.; Spiegelman, B.M.; Mortensen, R.M. PPARγ is required for the differentiation of adipose tissue in vivo and in vitro. Molecular cell 1999, 4, 611-617.

Comment 8

Line 99 – muslce8? Is it a reference?

Response

Thank you for your comment. We add the reference (Line 100).

Comment 9

Line 99-101 – please move the sentence on db/db mice to the next paragraph.

Response

Thank you for your comment. We move the sentence on db/db mice to the next paragraph (Line 101-102).

Comment 10

Line 102-111 – IO extract supplementation. How was it administered? In food supply or given orally via gavage?

Response

Thank you very much for your comment. Here we give more information about IO extract supplementation (Line 102-104).

Comment 11

Line 141 – “may be associated with the prevention” -> “may be effective as supplementary treatment of”

Response

Thank you for your comment. We replace the word to “may be effective as supplementary treatment of” (Line 144-145).

Comment 12

Line 158-159 – “these proteins are the downstream promoters of adipocyte-specific proteins, which are involved…” – “these transcription factors are highly expressed in adipocytes and are involved…”

Response

Thank you for your comment. We rewrite the sentence to “These transcription factors are highly expressed in adipocytes and are involved in the mediation of lipid synthesis, lipolysis, and glucose uptake in adipocytes.” (Line 161-162).

Comment 13

Line 175-176 – In food supply or given orally via gavage?

Response

Thank you for your comment. The explanations give above in the response for the comment 10.

Comment 14

Line 178-179 – move last sentence to the next paragraph.

Response

Thank you for your comment. We move last sentence to the next paragraph (Line 181-182).

Comment 15

Line 188 – “secretory responsiveness “of insulin”…”?

Response

Thank you for your comment. We add the word “of insulin” (Line 190).

Comment 16

Line 195 – “prevention” -> “management”

Response

Thank you for your comment. We replace the word to “management” (Line 197).

Comment 17

Line 101 – “prevent” -> “assist in the treatment of”

Response

Thank you for your comment. We replace the word to “assist in the treatment of” (Line 203).

Comment 18

Line 229-230 – “which supposedly have fewer”

Response

Thank you for your comment. We rewrite the sentence to “Therefore, researchers have been interested in identifying new anti-cancer drugs from marine sources, which supposedly have fewer adverse side effects unlike synthetic drugs [88].” (Line 231-233).

Comment 19

Line 232 – “caused by oxidative stress, which increases” -> “together with oxidative stress, which increase”

Response

Thank you for your comment. We rewrite the sentence to “Obesity is associated with lipid accumulation together with oxidative stress, which increase insulin resistance and eventually results in diabetes.” (Line 235-236).

Thank you for spending your most valuable time in evaluating our manuscript. We replace the word where the reviewer mentioned and address the additional information where needed. The alterations were marked in red color.

Kind Regards.

BoMi Ryu

Reviewer 2 Report

The manuscript is well reviewed about anti-obesity and anti-diabetic effects of Ishige Okamurae.

The authors should check the following points again:

1) Page 2, lines 81-82. The value of 109.8% was relative ROS production in the literature. IO  DID NOT inhibit ROS production.

2) Figure 1. Insurin resistance is down-regulated, not up-regulated. Inverse the direction of the arrow.

Author Response

Dear Academic Editor,

We are gratitude for your comments and spending your valuable time to read our manuscript. We have strongly considered your suggestions and have provided the answers to each comment. The revised contents are highlighted with red color in the revised manuscript.

Comment 1

Page 2, lines 81-82. The value of 109.8% was relative ROS production in the literature. IO DID NOT inhibit ROS production.

Response

Thank you for your comment. We agree/remove the sentence and clarify the free radical species that were inhibited by IO extract as below.

The antioxidant properties of Ishige foliacea and IO methanol extracts have been investigated in terms of their free-radical which include 1,1-diphenyl-2-picryl hydrazyl (DPPH), and 2,2-azobis(3-ethylbenzothiazoline-6-sulfonate (ABTS) and nitrite scavenging activity [21].” (Line 63-66)

Comment 2

Figure 1. Insurin resistance is down-regulated, not up-regulated. Inverse the direction of the arrow.

Response

We appreciate your clear observation. We clarify the Figure 1.

Thank you for spending your most valuable time in evaluating our manuscript. We replace the word where the reviewer mentioned and address the additional information where needed. The alterations were marked in red color.

Kind Regards.

BoMi Ryu
